# Effectiveness of Educational Interventions on Adherence to Lifestyle Modifications Among Hypertensive Patients: An Integrative Review

**DOI:** 10.3390/ijerph17072513

**Published:** 2020-04-07

**Authors:** Hon Lon Tam, Eliza Mi Ling Wong, Kin Cheung

**Affiliations:** 1School of Nursing, The Hong Kong Polytechnic University, Hong Kong, China; kin.cheung@polyu.edu.hk; 2Education department, Kiang Wu Nursing College of Macau, Macau 999078, China

**Keywords:** hypertension, health education, adherence, lifestyle

## Abstract

Controlling blood pressure is a global concern as it is a major risk factor for cardiometabolic diseases and stroke. A flattened control rate was noted in recent decades, which highlighted an issue of adherence to medications and lifestyle modifications. Effectiveness of educational intervention on medication adherence and blood pressure control had been reviewed, but reviews on lifestyle modifications are lacking. This review with meta-analysis aimed to identify the effect of educational interventions on blood pressure control and adherence to lifestyle modifications. In accordance with the PRISMA statement, a search of CINAHL Complete, PubMed, Medline, Embase and Scopus for randomized control trials published between 2009 and 2019 was conducted. Data were extracted for quality synthesis and meta-analysis. Thirteen studies were included. Two forms of educational intervention, individual and group education, were commonly used. Phone calls, message reminders and reading materials after education were identified in the studies as supportive methods, which showed a moderate to large effect on adherence to lifestyle modifications and blood pressure control. Monthly group education lasting 45 min was suggested. Health professionals could integrate the education with supportive methods into community health promotion to improve and reinforce the adherence behavior on medications and lifestyle modifications among hypertensive patients.

## 1. Introduction

Hypertension (HTN) is the leading risk factor causing cardiometabolic disease burden globally, and two of the complications, ischemic heart disease and stroke, contributed one-fourth of the global total deaths in 2016 [1,2,3,4]. A global collaboration on non-communicable disease (NCD) analyzed 1479 population-based studies from 1975 to 2015 and revealed that the global prevalence of HTN was in a decreasing trend over the decades, especially in high-income economies [5]. The NCD collaboration further analyzed 123 national surveys in 12 high-income economies from 1976 to 2017. The results show that the awareness, treatment and control rate of HTN were improved, but this has begun to flatten since 2005 [6]. Different interventions, such as yoga and guided breathing techniques, were claimed to have blood pressure (BP) lowering effects, but the results from a systematic review of randomized control trials (RCTs) were inconclusive [7,8]. They, therefore, were not suggested in the HTN management guidelines as effective interventions. Instead, the guidelines indicated that anti-hypertensive medications and lifestyle modifications should be combined in HTN management, in which lifestyle modifications in HTN seek to alter and maintain a recommended healthy habit that usually consists of five components: healthy diet, increased physical activity, weight control, smoking cessation, and limited alcohol consumption [9,10,11,12,13]. The NCD collaboration suggested health promotion was one of the keys to improving HTN management [6]. Health promotion should integrate the use of prescribed medications and lifestyle modifications, as they have been proven to control BP effectively [9,10,11,12,13]. With the unclear reason of the flatten control rate, a concern of adherence to HTN management was raised since approximately 60% of the treated hypertensive patients who did not achieve the target BP of systolic blood pressure (SBP) less than 140 mmHg and diastolic blood pressure (DBP) less than 90 mmHg [6,14].

Adherence to HTN management was defined as an individual’s behavior corresponding to the recommendations on the prescribed medications and lifestyle modifications [15]. A cross-sectional study of hypertensive patients in Turkey revealed a better medication adherence behavior, such that 78% of patients were reported to adhere to the prescribed anti-hypertensive medication, while the adherence to lifestyle modifications was only 42% [16]. A secondary analysis on an American national survey reported only 1.7% of hypertensive patients followed all five components of lifestyle modifications [17]. Low adherence to lifestyle modifications was highlighted among hypertensive patients, as the use of medication alone was not enough to control BP [9,10,11,12,13]. A need was raised to identify effective interventions to address the issue of low adherence to lifestyle modifications. In review of effective interventions to improve HTN management, educational interventions delivered to patients were identified to have significant effect on BP control and medication adherence in systematic reviews and meta-analysis [18,19,20]. However, there is a lack of reviews evaluating the effectiveness of educational interventions to improve the adherence to lifestyle modifications.

An integrative review with meta-analysis was conducted to summarize the evidence addressing the adherence to lifestyle modifications among hypertensive adults. The aim of this review was to identify the effect of educational interventions on adherence to lifestyle modifications for BP control, with specific focus on the effective components of the educational interventions in terms of delivery mode and dosage, theoretical framework and the use of any reinforcement measure or strategy after education as supportive methods.

## 2. Methods

### 2.1. Search Strategy

The review was conducted and reported in accordance with the Preferred Reporting Items for Systematic Reviews and Meta-Analyses (PRISMA) statement [21]. The framework from the PRISMA statement was used to guide the search, as below:
ParticipantsHypertensive adultsInterventionsEducational interventionsComparisonsStandard careOutcomesAdherence to lifestyle modifications, BPStudy designRCT

In the search, “hypertensi*” was set as the title with keywords “adult” AND “intervention OR program OR programme OR educat*” AND “lifestyle OR diet OR dietary OR exercise OR physical activity OR weight OR alcohol” AND “adherence OR compliance OR effect*”. The databases used in this search included Embase, Medline, PubMed, CINAHL Complete (via EbscoHost) and Scopus. These electronic databases were chosen as they are health sciences related and to cover as many studies as possible. In regard to the latest trend, publications in the most recent decade were searched for and the date of publications was set from 1st January 2009 to 31st December 2019. A manual search was conducted to identify additional potential eligible studies from the reference lists of eligible articles. All authors reviewed the search strategy and the search was conducted independently by two investigators.

### 2.2. Eligible Criteria and Quality Assessment

The eligible studies should fulfil the following criteria: (1) RCT; (2) educational intervention; (3) participants aged 18 or above; (4) participants diagnosed with HTN; (5) include content of the adherence to lifestyle modifications; (6) whole study period was less than 24 weeks; (7) written in English. Only RCTs were chosen to yield reliable evidence about the effectiveness of the interventions. All potentially relevant studies identified from the literature search were screened on the basis of title and abstract. Studies not meeting the above criteria and duplicated studies were excluded. Also, studies falling in any of the following criteria were excluded: (a) study on pregnancy; (b) kidney disease related HTN; (c) pulmonary HTN; (d) portal HTN; (e) intracranial HTN; (f) study of effect of medication or supplement; (g) study protocol or unpublished thesis. The remaining studies were assessed in full text to determine if they would be included in the review.

The Cochrane tool for assessing risk of bias in randomized trials was adopted for quality assessment by the first author and one doctorate student [22]. There were six parts in the tool to assess the risk of bias of articles. Any disagreement on study selection was discussed and resolved by the second author [22].

### 2.3. Data Synthesis and Meta-Analysis

A template of spreadsheet was developed for data extraction including study design (setting, sample size and attrition rate), characteristics of participants (age and gender), characteristics of educational interventions (delivery mode, theoretical framework, frequency, duration and supportive method) and outcome measures (adherence to lifestyle modifications and BP control). The data were extracted by the first author and compared based on the aim of the review, and the accuracy of data extraction was checked by the second author. As varied in time in each session of education among the studies, a weighted mean time was calculated by the sum of the product of time and sessions in each study then divided by the total number of sessions. The data of adherence to lifestyle modifications and BP were extracted for meta-analysis to examine the effectiveness of educational interventions.

Meta-analysis was conducted for adherence to lifestyle modifications and BP control by using Review Manager 5.3. Since a standard education was provided in the control group, a pre-post mean difference within the groups was used to assess differences between control and intervention groups. The data of intervention groups in the 3-arm RCTs were divided into two to compare with the control group, respectively. In line with the specific focus on the effective components of the educational interventions, three subgroup analyses, mode of education, theory-based intervention and the use of supportive methods after education, were performed to examine the effect on adherence to lifestyle modifications and BP control. Subgroup analysis was conducted if the comparator was more than one. Standardized mean difference (SMD) with a 95% confidence interval was used for the pooled effect of continuous variables. I^2^ statistic was performed to analyze heterogeneity [23]. Random-effects model was utilized if I^2^ > 50%. Otherwise, a fixed-effects model was used. A small effect size was noted if the SMD was equal to 0.2, 0.5 was a moderate effect, and a large effect was indicated by an SMD equal to 0.8 [24].

## 3. Results

A total of 5347 articles were initially identified from database searches and four were identified in a manual search (Figure 1). All the records were put into a web app, Rayyan, for duplicate removal and screening by two investigators [25]. There were 4120 articles screened for titles and abstracts, among which 60 articles were left for full-text and quality assessment. Thirteen RCTs satisfied the inclusion criteria and are described in detail in Appendix A. The characteristics of the studies, participants, educational interventions and outcome measures were summarized as follows.

### 3.1. Quality Assessment

The Cochrane risk-of-bias tool was updated in 2019, such that the six parts of a study were assessed namely: randomization process, intervention assignment, intervention adherence, missing outcome, outcome measurement, and result selection [22]. Figure 2 shows the quality of the included studies. The concealment of the randomized allocation was not mentioned clearly in some studies, which could increase the risk of bias in randomization process. Low risk of bias in intervention assignment was noted in several studies as they employed a similar protocol between control and intervention groups, or a fixed personnel was assigned to one group. The issue of intervention adherence was raised in a study in which 23.8% of participants in the intervention group did not follow the study protocol. The high refusal rate and baseline imbalance led to a high risk of bias in the missing outcome data [26]. Some concerns were raised in outcome measurement because the outcome was assessed by the interventionist or the blindness of the assessor to participant’s allocation was not mentioned clearly. All studies reported their results according to the pre-specified analysis plan. Overall, the included studies were kept for further data analysis after quality assessment.

### 3.2. Characteristics of Studies

The included studies were published from 2010 to 2019. According to the classification from The World Bank Group [27], five were from high-income economies [26,28,29,30,31] and eight from middle-income economies [32,33,34,35,36,37,38,39]. Of the 13 included studies, two were three-arm RCTs [31,32], and, in seven studies, their educational interventions were delivered by nurses [28,30,32,34,36,37,38].

All studies were conducted in a community or primary health setting, except one, which was conducted in hospital wards [38]. The end-point assessment was usually conducted in the same month of the last intervention delivered; only three studies stated the end-point assessment was conducted one month after the last intervention [30,33,37]. The sample size in each study ranged from 28 to 533, with an overall attrition rate from 0% to 22.7%. One study did not provide information on attrition rate [39].

### 3.3. Characteristics of Educational Interventions

The control group in most studies received standard education, which referred to the standard generalized HTN information formulated by the local healthcare setting. On the other hand, the intervention group received standard education with add-on information focusing on lifestyle modifications and support method, or the content of the education was revised from the local standardized HTN information. In brief, the education in the intervention groups consisted of lifestyle modifications in local context. Therefore, it was difficult to compare the content of the educational interventions between studies in different countries. In accordance with the specific focus of the review, the effective components in terms of delivery mode and dosage, theoretical framework and the use of supportive methods were described as follows.

#### 3.3.1. Delivery Mode and Dosage

Two modes of intervention delivery, individual education and group education were identified in the intervention groups. Individual education varied among HTN management counseling, home visit, and individualized lifestyle education. Their mode of delivery was in common; that is, it was in a one-on-one manner. The weighted mean time for an individual education session was 32.6 ± 10.1 min. Regarding the frequency, two studies provided only one individual education session during the whole study period [28,39]. Three studies provided regular individual education sessions in a monthly manner [29,31,32], while the others varied in frequency or were not mentioned clearly

For the group education, the number of participants in each group varied from two to 10 among the studies. Group education included health education and workshops with a weighted mean time of 42.3 ± 20.7 min. Monthly group education was commonly identified in the studies. It was noteworthy that Ribeiro et al. combined both individual and group education in their study, as further discussed in Section 4.1 [33].

#### 3.3.2. Theoretical Framework

A theoretical framework was highlighted, since eight out of thirteen studies used it to guide the study design. Among them, motivational interviewing and the stages of change model were commonly used. It was worth noting that these common theoretical frameworks were adopted in individual education, and only one study employed the stages of change model in group education [36]. Despite the study design, theoretical framework was also used to guide the development of reinforcement measures or strategies after education.

#### 3.3.3. Supportive Methods

Reinforcement measures or strategies after education were grouped as supportive methods for further analysis. Three supportive methods, i.e., phone calls, message reminders, and take-home reading materials, were identified in seven studies. A single method was employed in three studies, and four studies combined two supportive methods to reinforce the effect of education. Among the studies in which a phone call was used as the only supportive method, the frequency varied from biweekly to monthly. An educational leaflet was employed as the single supportive method in one study.

In regard to the combined method, Wan et al. adopted a self-developed handbook for participants to read at home, and weekly message reminder was used in line with the content of the handbook to reinforce the effect of education [38]. One study provided a cookbook and used biweekly phone calls as supportive methods, while others employed phone calls and message reminders in different frequency.

### 3.4. Meta-Analysis Results

#### 3.4.1. Adherence to Lifestyle Modifications

Various measures were used to assess the adherence on lifestyle modifications including self-developed scale and validated questionnaire. Of the included studies, three studies were not pooled because the data of adherence were provided in percentage or median [28,33,39]. Some studies assessed an overall adherence, while some studies measured only one component of lifestyle modifications. The meta-analysis results of overall adherence to lifestyle modifications, adherence to dietary recommendations, and adherence to physical activity recommendations are illustrated as follows.

Only four studies in individual education and one in group education provided data on overall adherence to lifestyle modifications. The subgroup analysis of supportive methods showed that the use of supportive methods magnified the overall adherence to lifestyle modifications significantly (SMD: 1.25 vs. 0.73, Figure 3).

Eight studies measuring the adherence to dietary recommendations were pooled for meta-analyses. Although insufficient significance was indicated, group education showed a large effect to improve participants’ adherence to dietary recommendations (Figure 4). The use of theory-based interventions reached a significant level, but the effect was not as good as those without a theoretical framework (SMD: 0.57 vs. 0.81, Figure 5). In turn, the use of supportive methods showed a similar effect as those without supportive methods on the adherence to dietary recommendations (Figure 6).

On the other hand, five studies measured the adherence to physical activity recommendations and the results show that the use of supportive methods was effective to enhance the adherence (Figure 7).

#### 3.4.2. Blood Pressure Control

Three studies were excluded in the analysis since the SBP or DBP was not presented [36,37,39]. Therefore, ten studies were pooled in the meta-analysis to examine the effect of lifestyle modification educational interventions on SBP and DBP. Theory-based interventions showed a moderate effect to improve SBP and DBP (Figure 8 and Figure 9). In addition to the theory-based interventions, the use of supportive methods had a moderate effect on BP control (Figure 10).

## 4. Discussion

This review highlighted the importance of educational intervention on adherence to lifestyle modifications as indicated in the meta-analysis and that a moderate effect in the adherence to dietary recommendations as well as physical activity recommendations was noted. A cohort study with four years follow-up in Finland revealed that hypertensive patients were less likely to adhere to lifestyle modifications after the initiation of medications, as evidence of an increased body weight and decreased level of physical activity [40]. This finding further supports the importance of educational intervention on lifestyle modifications in HTN management, especially on adherence to dietary and physical activity recommendations.

In addition to adherence behavior, lifestyle modification educational intervention also demonstrated a moderate effect on BP control, as shown in the meta-analysis. It was noteworthy that eight out of thirteen included studies were conducted in middle-income economies, which highlighted a concern of HTN management in this group. The peak value of global age-standardized mean SBP and DBP were shifted to low-income and middle-income economies [5], resulting in greater total deaths due to the complications of HTN in middle-income economies than high-income economies [1]. Medication cost was a major factor that influenced the HTN management in the world, but the data available from middle-income economies were scarce [41]. In turn, adherence to lifestyle modifications should be emphasized in HTN management as it was proven to have significant BP lowering effect and was the first-line treatment for hypertensive patients [9,10,11,12,13]. Lifestyle modification educational intervention should be employed.

Regarding the educational intervention, seven studies were nurse-led [28,30,32,34,36,37,38]. The role of nursing in health promotion has been advocated by the International Council of Nurses for decades [42]. A recent non-inferior RCT conducted in a community outpatient clinic revealed that nurse-led HTN management did not differ from physician consultation on BP control [43]. In addition, HTN is a NCD, meaning that that community-level action is important [2,44]. The World Health Organization emphasized the role of nursing in the community could make vital contributions to various population groups in the community [45]. With the key role of health promotion in HTN management, educational intervention conducted by nurses could potentially make major contribution to meet the needs of community. Hence, the effective components of lifestyle modification educational intervention in terms of delivery mode, use of theoretical framework and use of supportive methods were discussed as follows.

### 4.1. Delivery Mode

Individual education, group education and combined education were identified in the studies. The analyses of the effectiveness of these educations on adherence to lifestyle modifications for BP control were not conducted because of only one comparator. Only a subgroup analysis of adherence to dietary recommendations was conducted as it consisted of three studies providing group education (Figure 4). By comparing the weighted mean time between individual and group education, group education is more cost-effective, by which 10 min more were spent in each session to reach more hypertensive patients. Both the individual and group education shared a common characteristic, in that they were delivered in a monthly manner. As a result, group education might be a solution to improve the adherence to dietary recommendations.

In regard to the combined education, only a two-arm RCT with a small sample size (14 participants in each arm) was included in this review [33]. The study did not use any supportive methods that facilitated an exploration of the effect of combined education. Da Silva et al. conducted a two-arm RCT in Brazil and combined individual and group educations with the use of phone calls as a supportive method [46]. Each control arm and intervention arm consisted of 47 participants and the interventions lasted for 12 months. A large effect was noted in adherence to lifestyle modifications (0.83) and a moderate effect was noted in BP control (SBP: 0.51, DBP: 0.53). The effect of using supportive method on BP control in the 12-month study of da Silva et al. was similar to the subgroup analysis of the 6-month studies in this review. The study was unable to demonstrate that a longer period of intervention would produce a larger effect on adherence behavior [47]. Regardless of the whole study period, combined education might not produce a better effect on BP control than single education.

### 4.2. Use of Theoretical Framework

Most of the theoretical frameworks underpinned that the educational interventions aimed to provide health knowledge and enhance or motivate patients’ own health management by lifestyle modifications. The analysis of theory-based interventions showed less effectiveness on adherence behavior, which was contrary to previous meta-analysis on anti-hypertensive medication adherence [19]. However, a significant effect on BP control was noted. Among the theory-based studies, only two studies used supportive methods to enhance the effect of individual education. Various theories were used in these two studies, including Roy’s adaptation theory and health promotion model [32], and the health belief model [38]. On the other hand, motivational interviewing and the stages of change model were identified as common theoretical frameworks. A previous systematic review revealed that motivational interviewing might have small to moderate effect on medication adherence, while the effect of the stages of change model was small [19]. However, motivational interviewing might not be effective in BP control [20]. In brief, no theoretical framework was reviewed to reach statistical significance on BP control or medication adherence. Theoretical frameworks therefore need to be taken into account in further studies.

### 4.3. Use of Supportive Methods

This integrative review revealed that the use of supportive methods after educational intervention could improve the adherence to lifestyle modifications and BP control, which was in line with two systematic reviews on medication adherence in hypertensive patients [19,20]. A recent systematic review of six studies supported the use of message reminders in HTN management, but the frequency was inconclusive and the effect size could not be assessed because of insufficient data [48]. In addition to adherence to lifestyle modifications and BP control, the use of supportive methods lowered the attrition rate. The mean attrition rate of those studies using supportive methods was 7.21%, while those without supportive methods was 9.16%.

### 4.4. Limitations

There were some limitations of this review. Only English RCTs conducted from 2009 to 2019 were searched for and included. Grey literature, such as proceedings from conferences and unpublished thesis, was not searched for. The review included 13 studies for qualitative synthesis and meta-analysis. The findings should be interpreted with caution because of a small number of studies contributing to each of the subgroup analyses.

## 5. Implications

The above discussions provided advices for HTN management in community and insights for research. Health professionals, especially nurses, could conduct group education in community to improve and reinforce the adherence behavior among hypertensive patients. The education could be conducted in a monthly manner. A group size of four to six was suggested for easy management with a duration of 45 min in each session. Supportive methods, such as phone calls and message reminders, were effective in improving adherence behavior among hypertensive patients. Weekly message reminders and monthly phone calls were suggested. The content of education needs to be modified at the community level, since different regional needs were found in a country [49], but several components of lifestyle modifications should be covered, including healthy diet, increase physical activity, weight control, smoking cessation, and limit alcohol consumption.

Regarding the implications for research, the health promotion model was suggested to have a moderate effect on mediation adherence among hypertensive patients [19]. Future study could test the effect of the model on the adherence to lifestyle modifications. In order to make consistent comparisons between studies, existing measures for hypertensive patients such as Treatment Adherence Questionnaire for Patients with Hypertension (TAQPH) [50] and Hill-Bone Compliance to High Blood Pressure Therapy Scale (HB-HBP) [51] were suggested. Some scholars advised the use of TAQPH in research as it covered more aspects of adherence behavior [52], while HB-HBP had a variety of language choices [53]. The reliable and validated measures could also be used in practice to evaluate the outcome of educational interventions on HTN management.

Rather than the immediate effect of the interventions, the sustainable effect of the interventions on adherence behavior was also important. Three studies left an intervention-free period of a month to conduct the end-point assessment, and a significant difference was found between groups [30,33,37]. Future studies could examine the effectiveness of intervention in a longer intervention-free period, and identify a time to re-educate the hypertensive patients resulting in a persistent adherence to HTN management.

## 6. Conclusions

This integrative review adds more evidence for the effectiveness of educational intervention on lifestyle modifications and BP control for adults living in the community. The use of supportive methods, such as phone calls and message reminders, after education had a moderate to large effect on BP control and adherence to lifestyle modifications, as indicated in the meta-analysis. Theory-based educational intervention and consistent measures of adherence behavior are needed to be undertaken in future studies. Monthly group education can be conducted in the community to promote HTN management. In addition to the lifestyle modifications in education, client’s needs should be addressed in the local context. With the technology advancing at an exponential rate, the use of supportive methods incorporated in educational intervention is needed to catch up to present-day realities so as to further increase clients’ adherence behavior.

## Figures and Tables

**Figure 1 ijerph-17-02513-f001:**
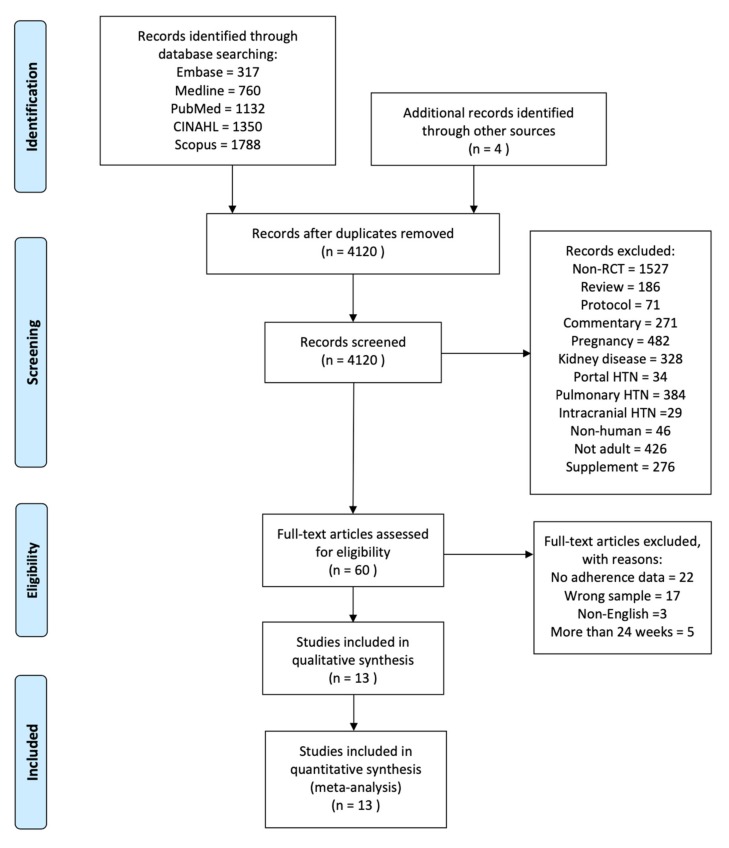
Flow diagram of literature selection process. (RCT = randomized control trial; HTN = hypertension).

**Figure 2 ijerph-17-02513-f002:**
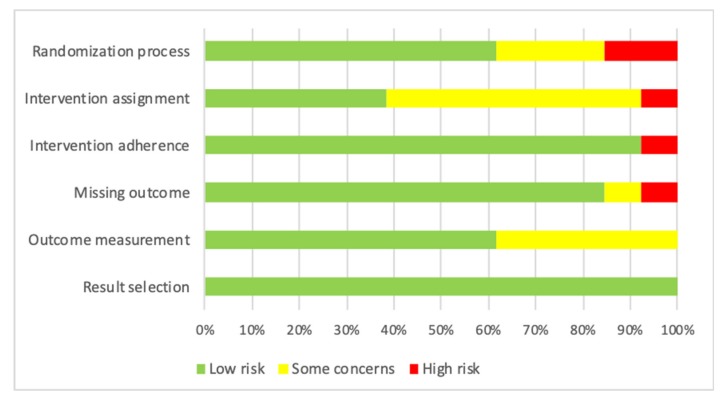
Risk of bias graph as percentage across all included studies.

**Figure 3 ijerph-17-02513-f003:**
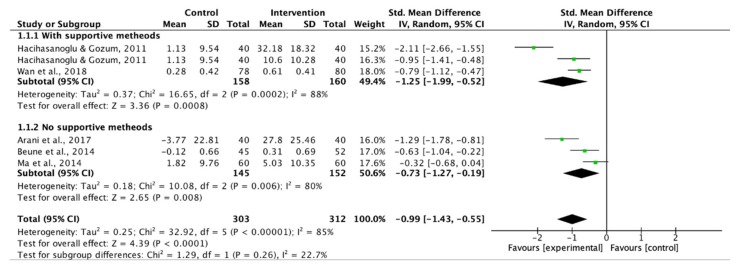
Forest plot of supportive methods on overall adherence to lifestyle modifications.

**Figure 4 ijerph-17-02513-f004:**
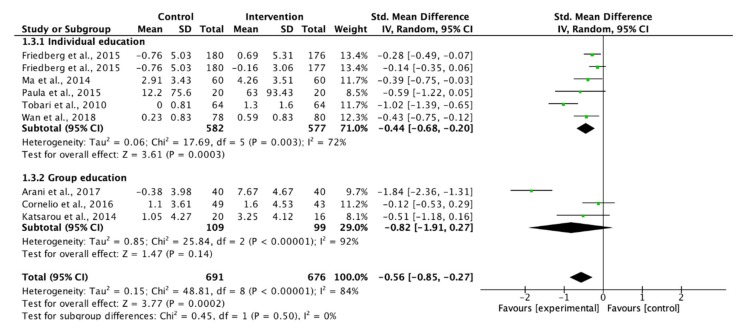
Forest plot of different educations on adherence to dietary recommendations.

**Figure 5 ijerph-17-02513-f005:**
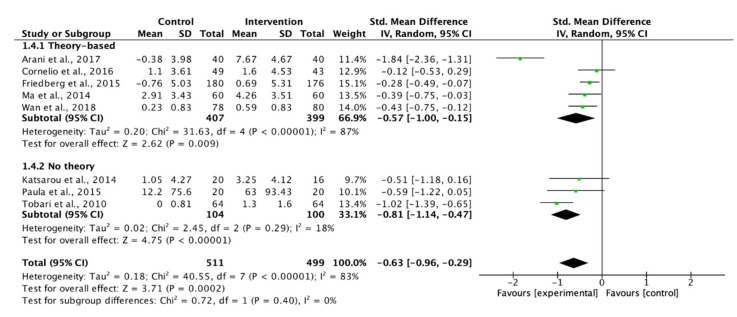
Forest plot of theory-based interventions on adherence to dietary recommendations.

**Figure 6 ijerph-17-02513-f006:**
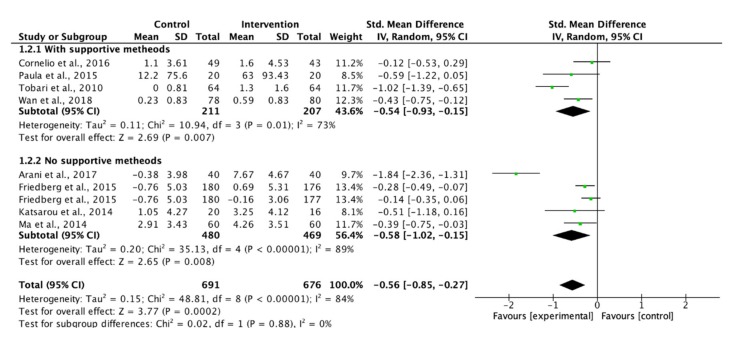
Forest plot of supportive methods on adherence to dietary recommendations.

**Figure 7 ijerph-17-02513-f007:**
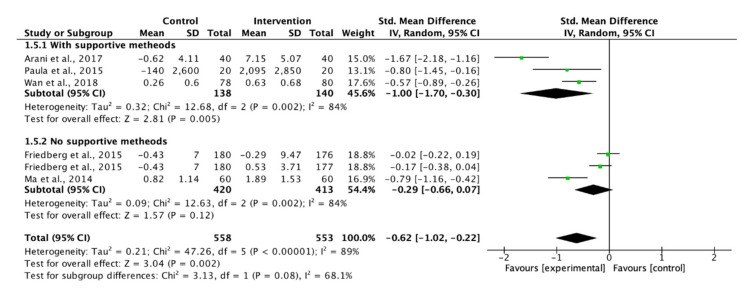
Forest plot of supportive methods on adherence to physical activity recommendations.

**Figure 8 ijerph-17-02513-f008:**
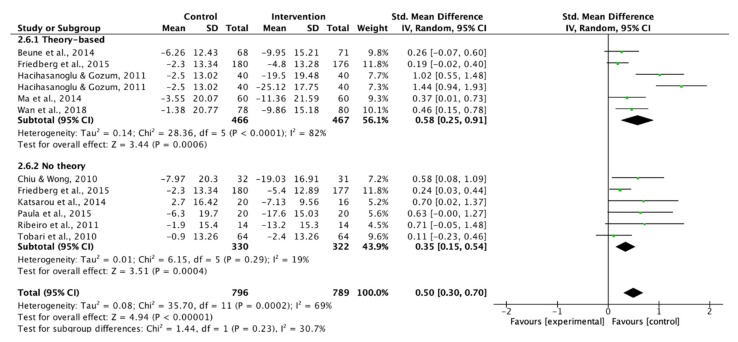
Forest plot of theory-based interventions on systolic blood pressure.

**Figure 9 ijerph-17-02513-f009:**
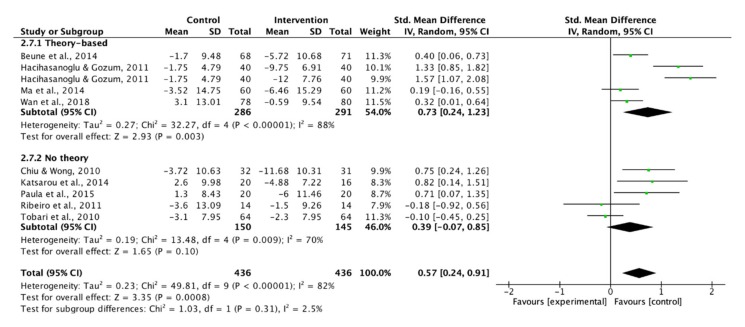
Forest plot of theory-based interventions on diastolic blood pressure.

**Figure 10 ijerph-17-02513-f010:**
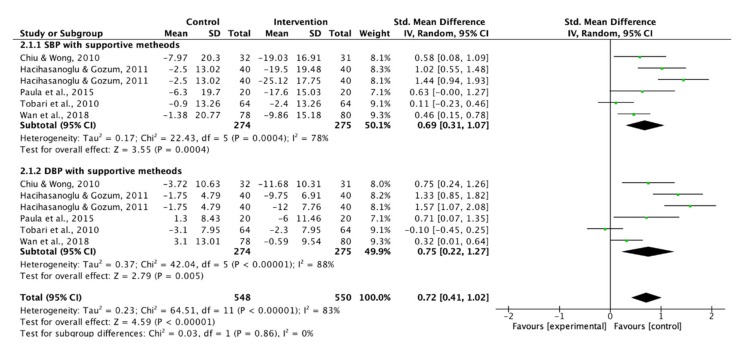
Forest plot of supportive methods on systolic blood pressure (SBP) and diastolic blood pressure (DBP).

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
