# Peer review of "Effectiveness of Educational Interventions on Adherence to Lifestyle Modifications Among Hypertensive Patients: An Integrative Review"

_ijerph, 2020, doi:10.3390/ijerph17072513_

Round 1

Reviewer 1 Report

Review  

Thank you for this opportunity to review this manuscript that aimed to understand and explain the effectiveness of educational interventions on adherence to lifestyle modifications among hypertensive patients as an integrative review. Also it is very interesting and is important as the metabolic disease tends to increase overall including hypertension.

The manuscript meets the aim and scope of the journal, but it needs to clarify as follows:

  1. Abstract

- You will need a brief description of the year and concept when searching for CINAHL Complete, PubMed, Medline, Embase and Scopus for a randomized controlled trial(It would be better to use what was described in the research purpose, results).

- After training, I think you should briefly and clearly present three strategies for strengthening measures.

- Does the use of the support method mean a moderate to large effect on lifestyle modifications and compliance with blood pressure control?

- I hope that you will improve the completion of the sentence with information on how to manage…. (ex: A 45-minute group education with a size of 4 to 6 was 23 suggested to conduct in a monthly manner.).

  1. Purpose of the study

The author should explain the purpose of the study in a logical, step-by-step, descriptive way to find direction for the need for the study.

  1. Why is hypertension an important factor in ischemic heart disease and stroke?
  2. High blood pressure is decreasing, but what are the factors suggested in the literature?

Why did flattening occur from 2005?

  1. Why is importance for the control of hypertension?
  2. What methods have been suggested in previous research literature to further reduce hypertension?
  3. How do you describe the concepts for lifestyle modification and education intervention?
  4. So why are you going to do this again in this study? what is the problems?

  1. Methods

The author used meta-analysis to understand the effects of lifestyle and educational interventions on hypertension. I think the methodology is suitable for this field of study, but supplement the explanation of the reason and process for selecting meta-analysis. (How to avoid potential bias, trials randomized?) . A detailed explanation of the analysis method is required. Moreover, please organize the table more briefly

  1. Results

In the conclusion part, there are too many subtitles, so the readability is poor, and it seems to be necessary to clearly sort the theorem with the table again to see if it is related to the research purpose. In addition, there is a need to organize the results of meta-analysis more clearly, without inserting references in the results section.

  1. Discussion of the findings

In the discussion, I would like to discuss the purpose of the research, the keywords to find through meta-analysis, and the results. The subtitles are related to the results, but there are still many subtitles themselves, making the study less consistent.

  1. Conclusions

The conclusions -section is a summary/repetition of the results, so the authors should rewrite and discuss the conclusions of the study.

I also suggest to consider what are the suggestions for the control the education and lifestyle for hypertension patients.

  1. Reference

 Organize references that fit the journal guide.

Author Response

Thank you for this opportunity to review this manuscript that aimed to understand and explain the effectiveness of educational interventions on adherence to lifestyle modifications among hypertensive patients as an integrative review. Also it is very interesting and is important as the metabolic disease tends to increase overall including hypertension.

Response: Thank you for the acknowledgement.

The manuscript meets the aim and scope of the journal, but it needs to clarify as follows:

  1. Abstract

- You will need a brief description of the year and concept when searching for CINAHL Complete, PubMed, Medline, Embase and Scopus for a randomized controlled trial(It would be better to use what was described in the research purpose, results).

Response: The year of publication and concept of search are added in the abstract. (Line 18-19)

- After training, I think you should briefly and clearly present three strategies for strengthening measures.

Response: The strategies are listed in the abstract as “Phone calls, message reminders and reading materials after education were identified in the studies as supportive methods”. (Line 22)

- Does the use of the support method mean a moderate to large effect on lifestyle modifications and compliance with blood pressure control?

Response: Amendments have been made as “Phone calls, message reminders and reading materials after education were identified in the studies as supportive methods, which showed a moderate to large effect on adherence to lifestyle modifications and blood pressure control.” (Line 22-23)

- I hope that you will improve the completion of the sentence with information on how to manage…. (ex: A 45-minute group education with a size of 4 to 6 was 23 suggested to conduct in a monthly manner.).

Response: The sentence is revised as “Monthly group education lasted 45 minutes was suggested. Health professionals could integrate the education with supportive methods into community health promotion to improve and reinforce the adherence behavior on medications and lifestyle modifications among hypertensive patients.” (Line 24-27)

  1. Purpose of the study

The author should explain the purpose of the study in a logical, step-by-step, descriptive way to find direction for the need for the study.

  1. Why is hypertension an important factor in ischemic heart disease and stroke? Response: “Hypertension (HTN) is the first leading risk factor to cause cardiometabolic disease burden globally, and two of the complications, ischemic heart disease and stroke, contributed one-fourth of the global total deaths in 2016.” (Line 33-35)

  1. High blood pressure is decreasing, but what are the factors suggested in the literature?

Response: “The non-communicable disease (NCD) collaboration suggested health promotion was one of the keys to improve the HTN management.” (Line 47-48)

Why did flattening occur from 2005?

Response: The exact cause was unclear. (Line 50)

  1. Why is importance for the control of hypertension?

Response: Additional information has been added in Introduction part as it is the first leading risk factor to cause cardiometabolic disease burden. (Line 33-35)

  1. What methods have been suggested in previous research literature to further reduce hypertension?

Response: Example of yoga and guided breathing technique are explored, but they are not included as effective interventions in the HTN management guidelines. Amendments have been made as “Different interventions such as yoga and guided breathing technique were claimed to have blood pressure (BP) lowering effect, but the results from systematic review of randomized control trials (RCTs) were inconclusive. They, therefore, were not suggested in the HTN management guidelines as effective interventions.” (Line 40-43)

  1. How do you describe the concepts for lifestyle modification and education intervention?

Response: “…Lifestyle modifications in HTN is to alter and maintain in a recommended healthy habit that usually consisted of five components: healthy diet, increase physical activity, weight control, smoking cessation, and limit alcohol consumption.” (Line45-47)  Educational intervention aims to provide health knowledge and enhance patient’s self-efficacy for self-help management. (Line 174-175, 318-320)

  1. So why are you going to do this again in this study? what is the problems?

Response: Lifestyle modifications are suggested in guidelines and should be combined with medication in HTN management. (Line 43-44)  But review of educational intervention on adherence to lifestyle modifications is lacking.

(Line 65-66)

  1. Methods

The author used meta-analysis to understand the effects of lifestyle and educational interventions on hypertension. I think the methodology is suitable for this field of study, but supplement the explanation of the reason and process for selecting meta-analysis. (How to avoid potential bias, trials randomized?) . A detailed explanation of the analysis method is required. Moreover, please organize the table more briefly.

Response: The method section is revised, especially the data synthesis and meta-analysis (section 2.3). The reason of subgroup analysis and the test of heterogeneity are added. (Line 123-126, 128-130).

  1. Results

In the conclusion part, there are too many subtitles, so the readability is poor, and it seems to be necessary to clearly sort the theorem with the table again to see if it is related to the research purpose. In addition, there is a need to organize the results of meta-analysis more clearly, without inserting references in the results section.

Response: Thanks for the suggestions. In review of the aim, several subtitles are revised and combined. (section 3.3.1)  Most of the references in the Results section were removed. We hope this will help readers follow the flow easily.

  1. Discussion of the findings

In the discussion, I would like to discuss the purpose of the research, the keywords to find through meta-analysis, and the results. The subtitles are related to the results, but there are still many subtitles themselves, making the study less consistent.

Response: Amendments have been made based on suggestions. (Line 265-283). The sequence of subtitles has been changed according to the aim of this review and hope this will help readers follow the flow easily.

  1. Conclusions

The conclusions -section is a summary/repetition of the results, so the authors should rewrite and discuss the conclusions of the study.

I also suggest to consider what are the suggestions for the control the education and lifestyle for hypertension patients.

Response: The conclusions section is revised to summarize the results of this review. Along with the suggestions, the mode of education and the content of lifestyle modifications are summarized in community-level for HTN management. (Line 374-383)

  1. Reference

 Organize references that fit the journal guide.

Response: The references are checked and revised accordingly.

Reviewer 2 Report

The article submitted for a review: Effectiveness of educational interventions on adherence to lifestyle modifications among hypertensive patients: An integrative review

discusses the important problem of determining the impact of educational interventions on compliance with lifestyle modifications to control blood pressure as a risk factor in ischemic heart disease and stroke. It contains an interestingly presented and prepared scientific meta-analysis of articles covering contemporary research in the aspect of educational activities in changing the lifestyle of people at risk of coronary heart disease. The article provides significant scientific support for the transmission of the results of these studies to the practice of work of health care professionals. It is also a great complement to the literature on the subject and can be considered for publication in this journal. The article was correctly formatted, divided into parts in accordance with the requirements of the magazine. It is written correctly and has appropriately selected, systematically analyzed, current literature. It is worth supporting the diligence of the publication before publishing by supplementing the content of the article with a few small things:

  • explaining to the reader why this scientific database data has been selected for analysis
  • arrangement of searched databases with increasing number of searched records (figure 1 / Identification)

Author Response

discusses the important problem of determining the impact of educational interventions on compliance with lifestyle modifications to control blood pressure as a risk factor in ischemic heart disease and stroke. It contains an interestingly presented and prepared scientific meta-analysis of articles covering contemporary research in the aspect of educational activities in changing the lifestyle of people at risk of coronary heart disease. The article provides significant scientific support for the transmission of the results of these studies to the practice of work of health care professionals. It is also a great complement to the literature on the subject and can be considered for publication in this journal. The article was correctly formatted, divided into parts in accordance with the requirements of the magazine. It is written correctly and has appropriately selected, systematically analyzed, current literature. It is worth supporting the diligence of the publication before publishing by supplementing the content of the article with a few small things:

Response: Thank you for the acknowledgement.

  • explaining to the reader why this scientific database data has been selected for analysis

Response: The chosen scientific databases are health sciences related; amendments have been made on Line 88-90.

  • arrangement of searched databases with increasing number of searched records (figure 1 / Identification).

Response: Figure 1 is revised for the arrangement of databases in an ascending order of number of searched records.

Round 2

Reviewer 1 Report

Dear Authors

I think you modified it very hard in my opinion.

Thank you.